# The Synergistic Mechanism of Total Saponins and Flavonoids in Notoginseng–Safflower against Myocardial Infarction Using a Comprehensive Metabolomics Strategy

**DOI:** 10.3390/molecules27248860

**Published:** 2022-12-13

**Authors:** Meng Fang, Yuqing Meng, Zhiyong Du, Mengqiu Guo, Yong Jiang, Pengfei Tu, Kun Hua, Yingyuan Lu, Xiaoyu Guo

**Affiliations:** 1State Key Laboratory of Natural and Biomimetic Drugs, School of Pharmaceutical Sciences, Peking University, Beijing 100191, China; 2Department of Cardiovascular Surgery, Beijing Anzhen Hospital, Capital Medical University, Beijing 100029, China

**Keywords:** notoginseng, safflower, synergistic, myocardial infarction, metabolomics

## Abstract

Notoginseng and safflower are commonly used traditional Chinese medicines for benefiting qi and activating blood circulation. A previous study by our group showed that the compatibility of the effective components of total saponins of notoginseng (NS) and total flavonoids of safflower (SF), named NS–SF, had a preventive effect on isoproterenol (ISO)-induced myocardial infarction (MI) in rats. However, the therapeutic effect on MI and the synergistic mechanism of NS–SF are still unclear. Therefore, integrated metabolomics, combined with immunohistochemistry and other pharmacological methods, was used to systematically research the therapeutic effect of NS–SF on MI rats and the synergistic mechanism of NS and SF. Compared to NS and SF, the results demonstrated that NS–SF exhibited a significantly better role in ameliorating myocardial damage, apoptosis, easing oxidative stress and anti-inflammation. NS–SF showed a more significant regulatory effect on metabolites involved in sphingolipid metabolism, glycine, serine, and threonine metabolism, primary bile acid biosynthesis, aminoacyl-tRNA biosynthesis, and tricarboxylic acid cycle, such as sphingosine, lysophosphatidylcholine (18:0), lysophosphatidylethanolamine (22:5/0:0), chenodeoxycholic acid, L-valine, glycine, and succinate, than NS or SF alone, indicating that NS and SF produced a synergistic effect on the treatment of MI. This study will provide a theoretical basis for the clinical development of NS–SF.

## 1. Introduction

Myocardial infarction (MI) is one of the most common traditional high-morbidity and lethal diseases, imposing a heavy burden on public health and society [1]. With the aging of society, changes in diet structure, and the influence of social and psychological factors, the incidence rate of MI has increased year by year [2]. MI is ischemic necrosis of the heart caused by sharply reduced or interrupted coronary blood flow [3]. Once MI exists for a long period of time, it can lead to sudden death or hemodynamic deterioration. The pathological mechanism and development of new therapeutic drugs for MI have become burning issues in the field of medical research [4].

Traditional Chinese medicines (TCMs) have been used to prevent and treat cardiovascular diseases for a long time [5]. *Panax notoginseng* (Burk.) F. H. Chen (Notoginseng) and *Carthamus tinctorius* L. (Safflower) in the form of an herb pair are commonly used in cardiovascular diseases. Total saponins are the main bioactive components in notoginseng (NS), and total flavonoids are the major bioactive ingredients in safflower (SF), which display protective effects against MI injury [6]. Previous studies have shown that the combination of NS and SF, named NS–SF, has a preventive effect on isoproterenol (ISO)-induced MI, and NS–SF was significantly better than NS and SF single drugs, which had a significant synergistic effect [7]. Compared with preventive drugs, it is more urgent to develop therapeutic drugs for MI in clinical practice, but the therapeutic effect on MI and the synergistic mechanism of NS–SF are still unclear. Therefore, it is necessary to explore the therapeutic effect of NS–SF on MI to provide new effective therapeutic drugs for the clinic.

Metabolomics, as a rapidly developed emerging discipline, has the ability to systematically and comprehensively analyze the changes in endogenous metabolites of biological systems stimulated by endogenous or exogenous sources [8]. It could also comprehensively explain the features of multiple ingredients, multiple targets, and multiple pathways of TCM [9]. Therefore, metabolomics has been widely used in the study of the efficacy mechanism of TCM compounds in recent years, which provides a new idea for the comprehensive and objective clarification of the overall mechanism of TCM [10,11]. The animal model of MI caused by the left anterior descending coronary artery (LADCA) was closest to the clinical features. In our study, the anti-MI effect of NS–SF was investigated utilizing a LADCA-induced rat model of MI. Pharmacological research methods combined with untargeted metabolomic methods based on nuclear magnetic resonance (NMR) and ultrahigh-performance liquid chromatography equipped with quadrupole time-of-flight mass spectrometry (UPLC-QTOF/MS) were employed to evaluate the effect of treatment and preliminarily reveal the therapeutic mechanism of NS–SF.

## 2. Results

### 2.1. Comparison of the Therapeutic Effects of NS, SF, and CNS in the MI Model

As shown in Figure 1, the diastolic and systolic functions of the left ventricle in the model group were decreased, and the EF and FS values were significantly decreased compared with rats in the normal and sham groups, indicating that the model was successfully constructed. The diastolic and systolic functions of the left ventricle were improved, and the EF and FS values were significantly increased in the NS, SF, NS–SF and positive groups (*p* < 0.05).

H&E staining and TUNEL assays provided evidence of the anti-MI effects of NS, SF, and NS–SF (Figure 2). The histopathological examination of the heart sections of MI rats stained with H&E showed severe inflammatory cell infiltration, cyanosis collagen deposition, disorder of myocardial cells, and blurring of myocardial fibers in the myocardial tissues of the MI group compared to the normal and sham groups. NS, SF and NS–SF reduced inflammatory cell infiltration and collagen deposition in MI rats. Compared with NS and SF, NS–SF had a more significant improvement effect, and NS–SF had the most obvious effect. The TUNEL assay indicated remarkable apoptosis in the myocardial tissues of the MI group compared with the sham group (Figure 2B,C). After administration of NS, SF, NS–SF and the positive drug, myocardial damage was ameliorated, as evidenced by decreased necrosis and TUNEL-positive cells. As displayed in Figure 2D, the levels of creatine kinase-myocardial band (CK-MB), cardiac troponin I (cTnI), lactate dehydrogenase (LDH), and aspartate transaminase (AST) were significantly increased in the model group compared with the normal and sham groups (*p* < 0.05). NS, SF and NS–SF inhibited the elevation of cTnI, CK-MB and LDH in plasma (*p* < 0.05). Compared with NS or SF alone, NS–SF can improve biochemical factors more significantly.

### 2.2. NS, SF, and CNS Inhibited Inflammatory Injury and Oxidative Stress Induced by MI

Inflammatory injury and oxidative injury are important pathological processes in MI. Tumor necrosis factor-α (TNF-α), interleukin 6 (IL-6), and IL-1α indexes related to inflammation in plasma were detected to judge the modeling situation and evaluate the strength of efficacy. NS, SF, NS–SF and the positive drug recovered myocardial inflammation induced by MI. The level of malondialdehyde (MDA), the end product of membrane lipid peroxidation in myocardial cells, was significantly increased, and the activities of superoxide dismutase (SOD) and glutathione peroxidase (GSH-Px), enzymes related to oxidative stress, were significantly decreased in MI rats. As shown in Figure 3, compared with the model group, the abnormal indexes of plasma oxidative stress recovered to different degrees in each administration group, and NS–SF displayed the best efficacy. These results confirmed that NS–SF demonstrated better activities in inhibiting myocardial cell injury than NS or SF alone.

### 2.3. NS, SF, and CNS Restored Global Metabolite Abnormalities in MI Rats

The representative chromatograms of the rat plasma samples by UPLC-QTOF/MS are displayed in Appendix A. UPLC-QTOF/MS-based metabolomic profiles collected under positive and negative ion modes and ^1^H NMR-based metabolomic profiles are shown in Figure 4. The results of cross validation demonstrated that R^2^ and Q^2^ were 0.614 and 0.508 in positive ion mode and 0.531 and 0.483 in negative ion mode, respectively, suggesting that the principal component analysis (PCA) model of UPLC-QTOF/MS data had good explanatory and predictive ability and could be used for multivariate statistical analysis (Figure 4A,B). Moreover, the cross-validation results showed that R^2^ and Q^2^ were 0.528 and 0.487, indicating that the PCA model of ^1^H NMR data had good explanatory and predictive ability (Figure 4C).

The results indicated that the plasma metabolic profile of model rats was significantly different from that of sham rats. The metabolic profiles of the NS–SF and positive groups were close to those of the normal and sham groups but far from those of the model group. After treatment with NS and SF, the metabolic profile of the NS and SF groups also showed a trend close to that of the sham groups, away from the model group, but the trend was not as obvious as that of the NS–SF group. The PCA score plot indicated that the recovery of the metabolic profile in MI rats by NS–SF was better than that by NS or SF alone, and the impact of NS–SF was equivalent to that of the positive drug.

### 2.4. Identification of Metabolic Alterations and Pathways Related to MI

The orthogonal partial least squares discriminant analysis (OPLS-DA) model was employed to analyze metabolic signatures in the sham and model groups (Figure 5 and Appendix A). The higher values of the R^2^X, R^2^Y and Q^2^ parameters of the model based on UPLC-QTOF/MS and ^1^H NMR-based metabolomic profiles suggested that the established OPLS-DA model had a good degree of interpretation and predictive ability, and without overfitting. In all, the OPLS-DA score plots demonstrated that the metabolic profiles of rats in the sham and model groups were significantly different, indicating the profile of endogenous metabolites in MI rats changed significantly. Quantities of 39 and 17 differential metabolites associated with MI were screened by UPLC/MS and ^1^H NMR, respectively (Table 1 and Table 2). The identified differential metabolites related to MI mainly involved a variety of metabolic pathways, including energy metabolism, bile acid metabolism, glycerophospholipid metabolism, fatty acid metabolism, and sphingolipid metabolism, suggesting that the MI model induced by left anterior descending coronary artery (LADCA) was closely related to the above metabolic disorders.

### 2.5. NS, SF, and NS–SF Showed Different Characteristics in Improving the Differential Metabolites Related to MI

As shown in Table 3, the differential metabolites were changed after administration of NS, SF, NS–SF, and the positive drug. NS, SF, and NS–SF could significantly regulate 25, 25, and 40 differential metabolites, respectively. Almost all of these metabolites regulated by NS and SF were significantly adjusted by NS–SF. Metabolites coregulated by NS, SF, and NS–SF were 3-oxocholic acid, ursocholic acid, margaric acid, uridine, taurocholic acid, sphinganine, phosphatidylethanolamine (PE) (19:0/0:0), PE (21:0/0:0), phosphatidylcholine (PC) (16:0/0:0), PC (18:0), L-leucine, lactate, threonine, and methylamine. Three metabolites (8-hydroxyguanosine, lysoPC (18:1), and trimethylamine) were uniquely regulated by NS. Phosphoglycolic acid and 3-hydroxyundecanoic acid were individually reversed by SF. Compared with NS and SF, NS–SF showed a more significant regulatory effect on metabolites involved in sphingolipid metabolism, glycine, serine and threonine metabolism, primary bile acid biosynthesis, aminoacyl-tRNA biosynthesis, and tricarboxylic acid cycle (TCA) cycle, such as sphingosine, sphingosine 1-phosphate, lysophosphatidylcholine (lysoPC) (18:0), lysophosphatidylethanolamine (lysoPE) (22:5/0:0), chenodeoxycholic acid, L-valine, glycine, creatine, and succinate (Appendix A). The metabolomics results gathered by UPLC/MS and ^1^H NMR suggested that the effect of NS–SF on MI was significantly better than that of NS or SF. Based on the information provided by the literature, HMDB, and KEGG databases, the screened differential metabolites were constructed into a disease network map (Figure 6).

## 3. Discussion

In this study, NS–SF had a therapeutic effect on MI. In terms of the pharmacodynamics results, the efficacy of NS–SF, including ameliorating myocardial damage, apoptosis, easing oxidative stress, and anti-inflammation, was significantly stronger than that of NS or SF alone. Moreover, UPLC/MS and ^1^H NMR metabolomic methods were applied to explore the therapeutic mechanisms and synergistic mechanisms of NS–SF against MI in rats. Studies have shown that NS–SF had a significantly stronger effect on MI than NS and SF. Compared with NS and SF, NS–SF could significantly improve the abnormality of energy metabolism, amino acid metabolism, glycerophospholipid metabolism, and bile acid metabolism caused by MI to better exert the therapeutic effect of NS–SF on MI rats, indicating that NS–SF had the characteristics of synergism.

The saponins of notoginseng could significantly reduce the myocardial infarction area of LADCA model rats, improve the left ventricular diastolic and systolic functions, and reduce the levels of creatine kinase and lactate dehydrogenase in serum, thus playing a role in myocardial protection [12]. Ginsenoside Rb1 and Rg1 were the main components of the saponins of notoginseng that lowered blood pressure by regulating phosphatidylinositol-3 kinase, protein serine/threonine kinase, endothelial nitric oxide synthase signal transduction pathway and the transport of L-arginine in endothelial cells, and increasing endothelial cell dependent vasodilation in rats [13]. Hydroxysafflor yellow A (HSYA), the main component of safflower, had a myocardial protective effect through reducing the area of myocardial infarction, increasing the activity of SOD, reducing the content of MDA, inhibiting the activity of endothelial nitric oxide synthase, and reducing the levels of NO and creatine kinase isoenzyme [14,15]. The pharmacological results demonstrated that NS–SF had a therapeutic effect through ameliorating myocardial damage, apoptosis, easing oxidative stress, and anti-inflammation in this research.

Many studies have shown abnormal energy metabolism in MI disease states [16]. Pyruvate was not only the product of the glycolysis pathway, but also of the carboxylation of pyruvate, which was an essential physiological process to maintain normal heart function [17]. The tricarboxylic acid cycle played a central role in oxidative phosphorylation of the myocardium, so under normal conditions, the contents of tricarboxylic acid intermediates such as malic acid and succinic acid were strictly regulated by the body [18]. In this study, compared with sham rats, the plasma contents of pyruvate, acetate, glucose, succinate, citrate, and malate, which are related to energy metabolism, were changed in MI rats, which suggested that MI mainly consumed glucose to obtain energy through glycolysis. The metabolites related to energy metabolism can be manipulated by NS, SF or NS–SF, and NS–SF could regulate succinate, citrate, and glucose compared with NS and malic acid compared with SF.

Previous studies have shown that the glutamate content was positively correlated with the size of MI [19]. Valine has been shown to reverse physiological changes caused by hypoxia and to provide cardiac protection during acute ischemia and hypoxia. Alanine was a nitrogen transporter produced by skeletal muscle [20]. Pyruvate could be converted into alanine through transamination to meet the energy requirements of some cardiomyocytes and was also the main energy supply pathway of cardiomyocytes [21]. L-tryptophan was closely associated with immune system activation and inflammation [22,23,24]. In this study, plasma glutamate was significantly increased in the model group, while the plasma glutamate, arginine, valine, alanine, and glutamine were significantly decreased, indicating an abnormal amino acid metabolic pathway in the model group. The NS–SF group was better than the NS group at regulating leucine and even more so at regulating valine and glycine. Compared with SF, NS–SF could better regulate valine and glycine. Methionine has been proven to improve cell oxidative balance and interact with various ROS, and the increase in methionine is related to the increase in SOD and GSH Px activity [25]. Thus, we speculated that NS–SF alleviated MI-induced oxidative stress through reducing the level of methionine. In addition, the increase in branched chain amino acids (BCAAs), including valine, leucine, and isoleucine, could lead to the accumulation of superoxide, thus promoting the progress of heart failure [26,27]. The results demonstrated that NS–SF could synergistically regulate the catabolism of impaired BCAAs in MI rats and reduce the level of MDA.

Endogenous metabolites, such as glycerolipids, sphingolipids, bile acids, and L-tryptophan, are closely related to inflammation and apoptosis [28]. In this study, the levels of lysoPCs in the model group were increased, while the levels of PCs, Pes, and lysoPEs were decreased, indicating that LADCA caused significant inflammation in the body. In addition, potential biomarker disorders associated with inflammation were further verified by H&E and TUNEL staining of myocardial tissue. Compared with the sham group, H&E staining in the model group showed serious infiltration of inflammatory factors in the myocardium, and TUNEL staining displayed significantly more apoptotic cells in the myocardium of the model group than in the sham group. Compared with NS and SF, NS–SF significantly inhibited the inflammatory damage of cardiomyocytes and reduced the level of inflammatory factors in circulation, suggesting that NS and SF played a synergistic anti-inflammatory role.

LysoPCs, mainly derived from PCs, could promote the inflammatory response and accelerate the progression of cardiovascular disease [29]. Disorder of the ratio of PCs to PEs could cause abnormal homeostatic membrane phospholipids [30]. LysoPEs were considered to be an important glycerolipid that caused arrhythmia [31,32]. The levels of lysoPEs were increased in the myocardium of MI rats caused by LADCA, suggesting that LADCA induced abnormal glycerolipid metabolism in rats. Compared with the NS and SF groups, NS–SF-regulated metabolites were connected to lipid metabolic pathways, such as lysoPC (18:0), lysoPE (22:5/0:0), PE (19:0/0:0), PC (16:0/0:0), PE (21:0/0:0), and PC (18:0). In addition, PCs were metabolized by intestinal microbes to produce choline and trimethylamine, which have been discovered to promote the development of atherosclerosis [33]. In MI rats, methylamine, dimethylamine, and trimethylamine were significantly increased. Compared to the sham group, NS, SF, and NS–SF significantly reduced methylamine levels, and NS–SF regulated dimethylamine more than SF.

Sphingolipids are highly bioactive compounds that played an important role in membrane structure, signaling cell proliferation, differentiation, apoptosis, and stress response [34,35]. Studies had revealed that sphingosinol-1-phosphoric acid could antagonize ceramide-induced apoptosis and effectively reverse heart ischemia reperfusion injury [36]. TNF-α-induced production and release of sphingosine could cause skeletal muscle atrophy by inducing apoptosis in muscle cells [37]. In this study, the levels of sphingosine in MI rats were significantly increased, while the level of sphingosine-1-phosphate was significantly decreased compared to the sham group, indicating that the body’s sphingosine metabolism was abnormal after MI, which further aggravated the degree of myocardial inflammation and apoptosis. NS–SF significantly reduced sphingosine content increased by MI compared with NS and SF. Moreover, sphingosine-1-phosphate was regulated only by NS–SF. These results demonstrated that NS–SF could significantly restore myocardial inflammation and apoptosis by regulating sphingomyelin metabolism and that NS and SF had a synergistic anti-inflammatory and anti-apoptotic effect.

Taurocholic acid, 3-oxocholic acid, ursocholic acid, cholic acid, deoxycholic acid, and chenodeoxycholic acid were all associated with bile acid metabolism [38]. Bile acids were the main components of bile and played a key role in maintaining fat metabolism [39]. Bile acids could promote the excretion, absorption and transport of fat, sterols and cholesterol in the intestinal tract and liver [40]. In this study, plasma bile acids, except ursolic acid, were significantly increased in MI rats compared with sham rats, indicating that MI could cause bile acid metabolism disorders and further aggravate the degree of MI. It is worth noting that all bile acid metabolites could be improved by NS–SF, and the improvement effect was stronger than that of NS or SF alone. These results indicated that the combination of NS and SF could significantly attenuate the disturbance of bile acid metabolism in MI rats to play a better anti-MI role.

## 4. Materials and Methods

### 4.1. Materials and Reagents

NS, the total saponins of notoginseng, was purchased from Yunnan Plant Pharmaceutical Co., Ltd. (Kunming, China) and the quality standard of NS complied with Chinese Pharmacopoeia (2010 edition), with notoginsenoside R1, ginsenoside Rg1, ginsenoside Re, ginsenoside Rb1, and ginsenoside Rd contents of 6.2%, 26.6%, 4.1%, 32.5% and 6.6%, respectively. Safflower was collected from Xinjiang Uygur Autonomous Region (China) and authenticated by one of the authors, Prof. Pengfei Tu, as the flowers of *Carthamus tinctorius* L. The voucher specimen of safflower (No. 20110301) was deposited at the Modern Research Center for Traditional Chinese Medicine, Peking University (Beijing, China). The reflux procedures of the materials of safflower were 120 L reflux for 1 h, 100 L reflux for 0.5 h, and 100 L reflux for 0.5 h at 80 °C using deionized water, and then the extract was filtered and concentrated in vacuo. The concentrated solution was subjected to a D101 macroporous resin column eluted with deionized water and 50% aqueous ethanol. SF was obtained by drying 50% aqueous ethanol eluate through spray drying. The levels of the effective components (total safflor yellow and total flavonoids) in SF were more than 40% using ultraviolet-visible spectrophotometry. Meanwhile, the content of HSYA and kaempferol-3-O-rutinoside in SF were more than 8.0% and 0.20% by high performance liquid chromatography, respectively [12]. The ratio of NS and SF was 6:5 in NS–SF.

The H&E staining kit was purchased from Beijing Legian Biotechnology Co., Ltd. (Beijing, China). The TUNEL kit was provided by Roche Co., Ltd. (Basel, Switzerland). The Cardiac troponin I (cTnI) ELISA kit was purchased from Wuhan Huamei Biological Engineering Co., Ltd. (Wuhan, China). The CK-MB, LDH and AST assay kits, SOD and MDA ELISA kits were supplied by Jiancheng Institute of Biotechnology (Nanjing, China). The GSH-Px kit was purchased from Wuhan Xinqidi Biotechnology Co., Ltd. (Wuhan, China).

Acetonitrile (LC-MS grade) and methanol (LC-MS grade) were acquired from Merck (Darmstadt, Germany). HPLC grade formic acid was provided by Fisher Scientific (Pittsburg, PA, USA). Ultrapure water (18.2 MΩ) was prepared using a Milli-Q water purification system (Millipore, MA, USA). Other chemicals were of analytical grade. Ursocholic acid, cholic acid, deoxycholic acid, phosphoglycolic acid, succinate, malic acid, and taurocholic acid were obtained from Beijing Bailingwei Technology Co., Ltd. (Beijing, China). 8-Hydroxyguanosine and 8-hydroxyguanine were purchased from Abcam Co., Ltd. (Cambridge, Britain). 3-(Trimethylsilyl) propionic-2,2,3,3-*d*_4_ acid sodium salt (TSP) was supplied from Isotope Laboratories Inc. reagent company (Cambridge, MA, USA). The purities of all standard products were above 98%. Diltiazem hydrochloride tablets were purchased from Asia-Pacific Pharmaceutical Co., Ltd. (Shaoxing, Zhejiang, China). Furosemide injection and lidocaine hydrochloride injection were provided by Hubei Tianyao Pharmaceutical Co., Ltd. (Xiangyang, Hubei, China).

### 4.2. Animals

Male Sprague–Dawley rats (215 ± 10 g) were obtained from the Laboratory Animal Center, Peking University Medical Department (Beijing, China). The temperature and humidity were set at 20 ± 2 °C and 70%, respectively. Rats were fed a certified standard diet and water in a 12 h/12 h light and dark cycle. They were allowed 3 days to adapt the laboratory environment before the experiment. All procedures followed the relevant national legislation and were approved by the Institutional Animal Care and Use Committee of Peking University Health Science Center (No. LA2015061).

### 4.3. MI Model and Drug Administration

The MI model was induced by LADCA following a method previously described in the literature [41,42]. The experiment animals were randomly divided into seven groups (6 rats per group): normal group, sham group, model group, CNS group, NS group, SF group, and positive group. Rats in the normal group were not operated on and rats in the sham group were not ligated. The other groups underwent ligation. The positive group is a positive control group. Based on previous research, the CNS group, NS group, SF group, and positive group were treated with CNS (NS:SF = 6:5, 55.0 mg/kg/d), NS (30.0 mg/kg/d), SF (25.0 mg/kg/d), and diltiazem hydrochloride (30.0 mg/kg/d) once a day for 7 days after surgery, respectively [43].

### 4.4. Echocardiographic Evaluation

The M-mode of the Vevo 2011 imaging system was used to record the left ventricular motion curve after drug administration for 7 consecutive days. The sampling frequency and scanning speed were set at 1000 times/s and 50–100 mm/s, respectively. The left ventricular internal diameter end of left ventricular posterior wall during systole (LVPWs), left ventricular posterior wall during diastole (LVPWd), left ventricular anterior wall during systole (LVAWs), left ventricular anterior wall during diastole (LVAWd), left ventricular internal diameter end systole (LVIDs), left ventricular internal diameter end diastole (LVIDd), ejection fraction (EF), and fractional shortening (FS) were recorded, and the data were averaged over three consecutive cardiac cycles.

### 4.5. Sample Collection

Blood samples were collected from the abdominal aorta and centrifuged for 10 min at 3500 rpm, 4 °C. The heart was quickly removed and rinsed with cold saline. One part of the heart was fixed in 4% paraformaldehyde for 24 h for tissue section experiments and the other was stored in −80 °C.

### 4.6. Histological Examination

The heart tissues were fixed with 4% paraformaldehyde for at least 24 h. After routine dehydration, paraffin embedding, and sectioning, the heart tissue specimens were stained with H&E staining for histological examination. TUNEL staining of the heart tissues was performed to determine the apoptotic cardiomyocytes.

### 4.7. Biochemical Indicators

cTnI was determined using an ELISA kit according to Huamei’s instructions (Huamei Biological Engineering Co., Ltd., Wuhan, China). The activity of GSH-Px in heart tissues was examined with detection kits following Wuhan Xinqidi Biotechnology’s protocol (Wuhan, China). The concentrations of CK-MB, LDH, AST, SOD, and MDA in serum samples were measured following the instructions of commercial kits (Jiancheng Co., Ltd., Nanjing, China). CK-MB, cTnI, LDH, and AST are the diagnostic indicators of myocardial infarction, and GSH-Px, SOD, and MDA are expressed as oxidative pathogenesis indicators in the present study [44,45].

### 4.8. Sample Preparation for UPLC-QTOF/MS Analysis

A volume of 50 μL of plasma was added to 150 μL of ice-cold solution (methanol: acetonitrile = 1:1, *v*/*v*) containing 0.1 mg/mL L-phenyl-d5-alanine and 0.1 mg/mL lysoPC (19:0) as the internal standards to precipitate the proteins. After vortexing for 1 min, the mixture was centrifuged at 14,000 rpm for 10 min at 4 °C. The supernatant was concentrated in a vacuum centrifugal concentrator. The dried residue was reconstituted in 80 μL ice-cold solvents (water:methanol:acetonitrile = 10:3:3, *v*/*v*/*v*) and centrifuged at 14,000 rpm for 5 min at 4 °C. The supernatant was collected for UPLC-QTOF/MS analysis.

All metabolomic analysis was performed on an UPLC-SYNAPT Xevo-G2 XS Q-TOFMS system (Waters Corp., Milford, CT, USA). The separations of plasma samples were performed on an ACQUITY UPLC BEH C18 column (2.1 mm × 100 mm, 1.7 μm, Waters Corp., Milford, CT, USA). The column was maintained at 40 °C, and the flow rate was 0.4 mL/min as a 1 μL aliquot of each sample was injected. The optimal mobile phase consisted of a linear gradient system of (A) 0.1% formic acid in water and (B) 0.1% formic acid in acetonitrile: 0–6.0 min, 3–60% B; 6.0–10.0 min, 60–90% B; 10.0–12.0 min, 90–100% B; 12.0–13.0 min, 100–3% B; 13–15.5 min, 3% B. The MS parameters were as follows: both positive and negative ion modes were applied; the source temperature was set at 115 °C; the desolvation gas temperature and desolvation gas flow were 500 °C and 800 L/h, respectively; the capillary voltage was 3.0 kV for positive ion mode and 2.2 kV for negative ion model; the sampling cone voltage was 35 V; the cone gas rate was set at 50 L/h.

UPLC-QTOF/MS data were processed by Progenesis QI (version 2.0, Waters, Milford, CT, USA) for peak alignment, grouping, peak extraction, and peak identification operations. Metabolites were identified through the primary and secondary mass spectrometry fragment information provided by the online databases of HMDB and LIPIDMAPS.

### 4.9. Sample Preparation for NMR Analysis

A volume of 300 μL plasma was added to 100 μL of 0.9% saline containing 0.1% TSP and 200 μL of D_2_O. After vortex-blending for 30 s, the mixture was centrifuged at 12,000 rpm for 10 min at 4 °C. Then, a volume of 450 μL of supernatant was added to a 5 mm sample tube after vertexing and mixing for NMR analysis.

All ^1^H NMR data were obtained by the relaxation editing pulse sequence (Carr-Purcell-Meiboom-Gill, CPMG) called on the Varian VNMRS-500 MHz nuclear magnetic resonance at 28 °C. The parameters were set as follows: the spectral width was 10,000 Hz; the mixing time was 0.10 s; the relaxation time was 4 s; the number of sampling points was 64 k; the number of accumulation times was 64; the time of each scan was 1.548 s; the experimental temperature was 320 K; the presaturation method was used to suppress the water peak.

^1^H NMR spectra were imported into MestreNova software (version 11, MestreLab Research, Santiago de Compostela, Spain) to adjust the baseline and phase. The TSP was selected as the location of the chemical shift reference peak, which was set as 0.00 ppm. After phase and baseline adjustments, spectral peaks in the range of δ 0.5–4.5 ppm were selected, and fractional integration was carried out according to 0.04 ppm for each segment. The integral was normalized according to the total integral strength of each spectrum. The obtained datasets were then used for statistical analysis.

### 4.10. Statistical Analysis

The obtained datasets for LC-MS and ^1^H NMR were imported into SIMCA˗P (v 14.1; Umetrics, Umea, Sweden) for multivariate statistical analysis. The PCA model was applied to determine the overall clustering trend, and the OPLS-DA model was used for multivariate statistical analysis. The Pareto variance (Par) scaling method was used. A combination of the *S*-plot (an absolute *p* (corr) > 0.4 was used as the cutoff value) and the variable influence in the projection (VIP) plot (VIP > 1.5) from the OPLS-DA model was performed to identify the differential metabolites for plasma UPLC-QTOF/MS data and ^1^H NMR mapping data. Furthermore, Student’s *t*-test (*p* < 0.05) and the univariate receiver operating characteristic (ROC) curve, using the area under the ROC curve (AUC ≥ 0.8), were used to evaluate the accuracy of these differential metabolites.

The non-metabolomic data were expressed as the mean ± standard deviation (SD). One-way analyses of variance (ANOVA) were performed using a Bonferroni correction of the GraphPad Prism 6 software (GraphPad Software, San Diego, CA, USA). A critical *p* value of <0.05 was considered statistically significant.

## 5. Conclusions

In summary, NS–SF had synergistic effects on MI by improving inflammation and myocardial cell apoptosis, and regulating energy metabolism, amino acid metabolism, glycerin, and abnormal bile acid and phospholipid metabolism. This research explained the synergistic mechanism of NS–SF against MI from a metabolomics perspective and provided a theoretical basis for the clinical application of NS–SF.

## Figures and Tables

**Figure 1 molecules-27-08860-f001:**
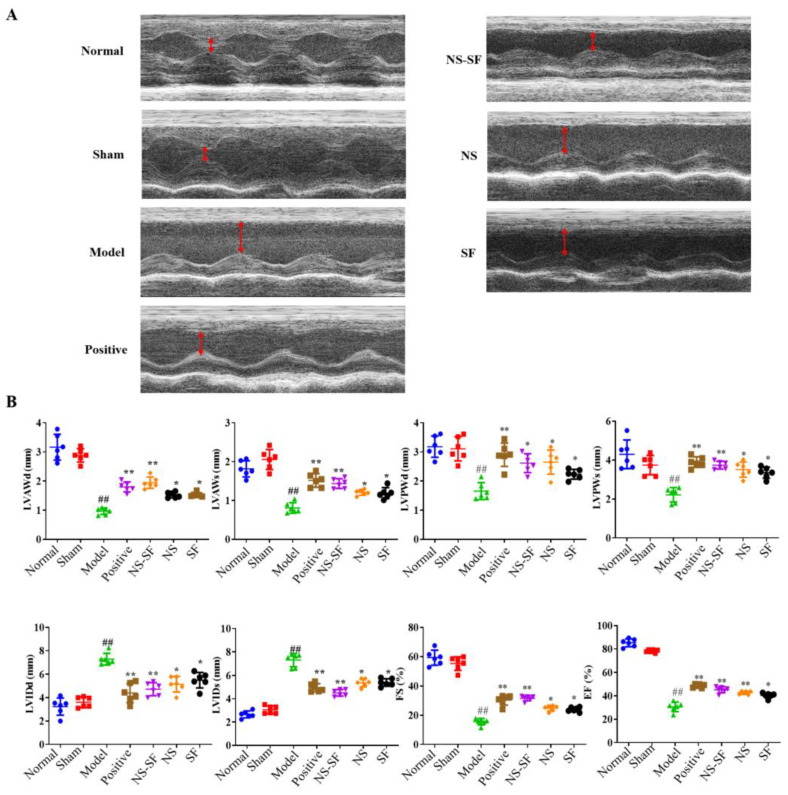
Evaluation of the heart functions of rats in every group. (**A**) The M-mode echocardiograms for rats in different groups. (**B**) The parameters of ECG-M for rats in different groups (mean ± SD, *n* = 6). ^##^
*p* < 0.01, when versus sham group; * *p* < 0.05, ** *p* < 0.01, when versus model group. Normal: normal group; Sham: sham-operated group; Model: model group; Positive group: positive control group; NS: total saponins of notoginseng group; SF: total flavonoids of safflower group; NS–SF: the compatibility of NS and SF group.

**Figure 2 molecules-27-08860-f002:**
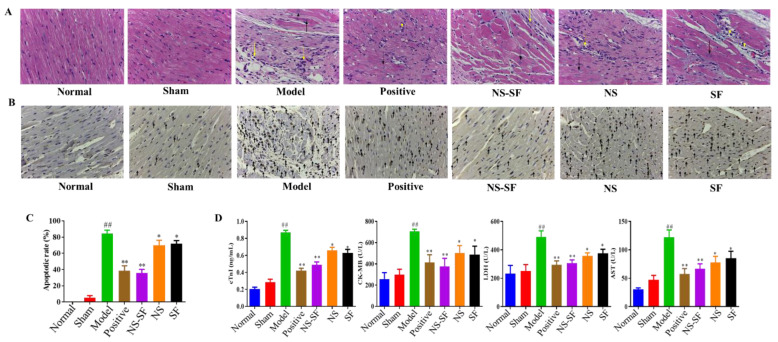
Effects of NS, SF and NS–SF on myocardial fibrosis and apoptosis induced by MI in rat heart tissues. (**A**) H&E staining of left ventricular (LV) tissue showed pathological and morphological changes in different groups (magnification, ×400). The necrotic myocardium is labeled with a black arrow, and inflammatory cells are labeled with a yellow arrow. (**B**) The results of TUNEL staining in different groups. Arrows indicate apoptotic cardiomyocyte nuclei. (**C**) Apoptosis rate of myocardial cells in each group (mean ± SD, *n* = 6). (**D**) The serum levels of MI-associated biochemical markers of rats in different groups (mean ± SD, *n* = 6). ^##^
*p* < 0.01, when versus sham group; * *p* < 0.05, ** *p* < 0.01, when versus model group. Normal: normal group; Sham: sham-operated group; Model: model group; Positive group: positive control group; NS: total saponins of notoginseng group; SF: total flavonoids of safflower group; NS–SF: the compatibility of NS and SF group.

**Figure 3 molecules-27-08860-f003:**
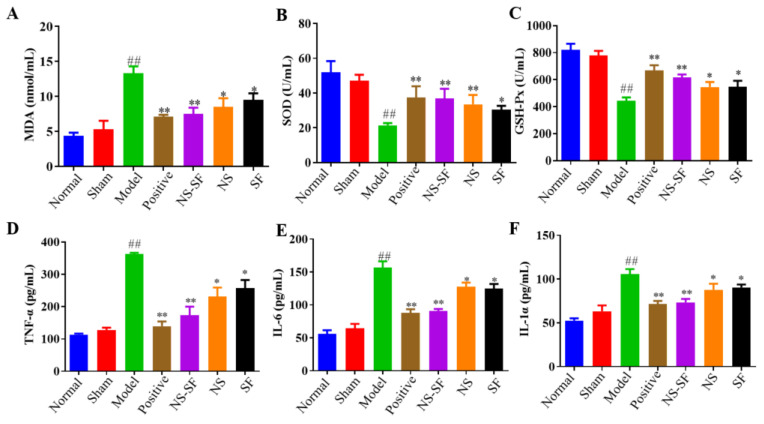
Effects of NS, SF and NS–SF on plasma anti-inflammatory and antioxidant activities of MI-induced cardiac hypertrophy in rats (mean ± SD, *n* = 6), (**A**) MDA, (**B**) SOD, (**C**) GSH-Px, (**D**) TNF-α, (**E**) IL-6, and (**F**) IL-1α. ^##^
*p* < 0.01, when versus sham group; *p* < 0.05, * *p* < 0.01, when versus model group. Normal: normal group; Sham: sham-operated group; Model: model group; Positive group: positive control group; NS: total saponins of notoginseng group; SF: total flavonoids of safflower group; NS–SF: the compatibility of NS and SF group.

**Figure 4 molecules-27-08860-f004:**
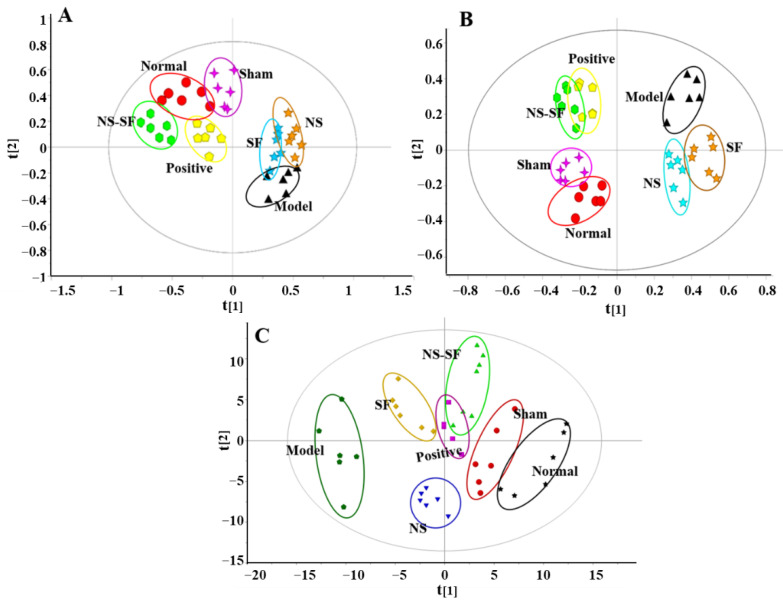
The PCA score plot of rats in different groups. (**A**) Positive mode by UPLC−QTOF/MS. (**B**) Negative mode by UPLC−QTOF/MS (**C**) ^1^H NMR. Normal: normal group; Sham: sham−operated group; Model: model group; Positive group: positive control group; NS: total saponins of notoginseng group; SF: total flavonoids of safflower group; NS–SF: the compatibility of NS and SF group.

**Figure 5 molecules-27-08860-f005:**
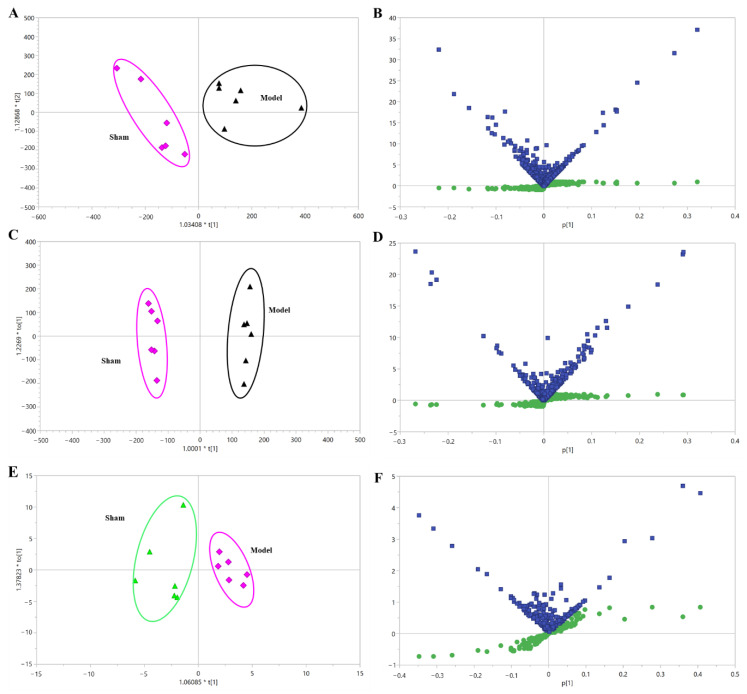
The OPLS-DA of the plasma samples from sham and model rats. (**A**) Positive mode by UPLC-QTOF/MS. (**B**) A combination plot of S-plot and VIP values in positive mode. (**C**) Negative mode by UPLC-QTOF/MS. (**D**) A combination plot of S-plot and VIP values in negative mode. (**E**) ^1^H NMR. (**F**) A combination plot of S-plot (green squares) and VIP (blue squares) values in ^1^H NMR. Sham: sham-operated group; Model: model group.

**Figure 6 molecules-27-08860-f006:**
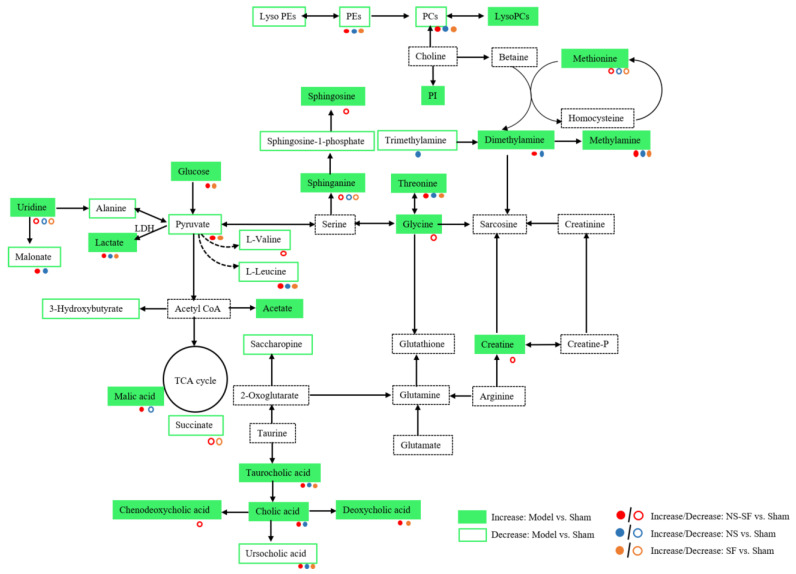
Schematic representation of the effects of NS, SF, and NS–SF on the metabolic network of the vital MI-altered metabolites. Sham: sham-operated group; Model: model group; Positive group: positive control group; NS: total saponins of notoginseng group; SF: total flavonoids of safflower group; NS–SF: the compatibility of NS and SF group.

**Table 1 molecules-27-08860-t001:** Identification of the potential biomarkers in plasma of the MI rats based on UPLC-QTOF/MS.

			Detected					
No.	t_R_	*m*/*z*	Formula	Mode	Identification	HMDB IDs	Trend	Pathway
1	0.75	299.2401	C_10_H_13_N_5_O_6_	–	8-Hydroxyguanosine	HMDB0002044	↑	Metabolism of nucleotides
2	0.83	192.1235	C_6_H_8_O_7_	–	Citrate	HMDB0000094	↓	TCA cycle
3	5.41	406.5555	C_24_H_38_O_5_	–	3-oxocholic acid	HMDB0000502	↑	Bile acid metabolism
4	5.42	408.5714	C_24_H_40_O_5_	–	Ursocholic acid	HMDB0000917	↓	Secondary bile acid biosynthesis
5	5.42	408.2875	C_24_H_40_O_5_	–	Cholic acid	HMDB0000619	↓	Primary bile acid biosynthesis
6	7.42	392.2926	C_24_H_40_O_4_	–	Deoxycholic acid	HMDB0000626	↓	Secondary bile acid biosynthesis
7	10.01	155.9823	C_2_H_5_O_6_P	–	Phosphoglycolic acid	HMDB0000816	↑	Glyoxylate and dicarboxylatemetabolism
8	10.99	118.0266	C_4_H_6_O_4_	–	Succinate	HMDB0000254	↓	TCA cycle
9	11.66	292.2038	C_18_H_28_O_3_	–	alpha-Licanic acid	LMFA02000194	↓	Fatty acid metabolism
10	14.02	134.0874	C_4_H_6_O_5_	–	Malic acid	HMDB0000744	↑	TCA cycle
11	0.57	233.2616	C_10_H_19_NO_5_	+	Hydroxypropionylcarnitine	HMDB0013125	↑	Fat metabolism
12	0.64	202.2906	C_11_H_22_O_3_	+	3-hydroxyundecanoic acid	HMDB0061654	↑	Fatty acid metabolism
13	1.62	276.2863	C_11_H_20_N_2_O_6_	+	Saccharopine	HMDB0000279	↓	Lysine degradation
14	2.67	650.2801	C_25_H_47_O_12_P	+	PI (16:1/0:0)	LMGF06050009	↑	Glycerophospholipid metabolism
15	6.21	273.2744	C_17_H_31_D_3_O_2_	+	Margaric acid	LMFA01010048	↓	Fatty acid metabolism
16	6.84	167.1255	C_5_H_5_N_5_O_2_	+	8-hydroxyguanine	HMDB0002032	↑	Purines and purine derivatives
17	7.61	541.3168	C_28_H_48_NO_7_P	+	LysoPC (20:5)	HMDB0010397	↑	Glycerophospholipid metabolism
18	8.32	301.5078	C_18_H_39_NO_2_	+	Sphinganine	HMDB0000269	↑	Sphingolipid metabolism
19	8.53	379.4718	C_18_H_38_NO_5_P	+	Sphigosine-1-phosphate	HMDB0000277	↓	Sphingolipid metabolism
20	8.86	523.6832	C_26_H_54_NO_7_P	+	LysoPC (18:0)	HMDB0010384	↑	Glycerophospholipid metabolism
21	8.87	299.4919	C_18_H_37_NO_2_	+	Sphingosine	HMDB0000252	↓	Sphingolipid metabolism
22	8.87	495.3325	C_24_H_50_NO_7_P	+	PE (19:0/0:0)	LMGP02050028	↓	Glycerophospholipid metabolism
23	9.01	317.2202	C_16_H_31_NO_5_	+	3-hydroxynonanoyl carnitine	HMDB0061635	↓	Fat metabolism
24	9.33	392.2926	C_24_H_40_O_4_	+	Chenodeoxycholic acid	HMDB0000518	↑	Primary bile acid biosynthesis
25	9.37	495.3325	C_24_H_50_NO_7_P	+	PC (16:0/0:0)	LMGP01050018	↓	Glycerophospholipid metabolism
26	9.39	527.6304	C_27_H_46_NO_7_P	+	LysoPE(22:5/0:0)	HMDB0011524	↓	Glycerophospholipid metabolism
27	9.52	521.3481	C_26_H_52_NO_7_P	+	PC (18:1/0:0)	LMGP01050029	↓	Glycerophospholipid metabolism
28	9.52	521.3481	C_26_H_52_NO_7_P	+	LysoPC (18:1)	HMDB0002815	↑	Glycerophospholipid metabolism
29	10.01	375.5878	C_24_H_41_NO_2_	+	Adrenoyl ethanolamide	HMDB0013626	↑	Fatty acid metabolism
30	10.21	244.2014	C_9_H_12_N_2_O_6_	+	Uridine	HMDB0000296	↑	Pyrimidine metabolism
31	10.69	515.2916	C_26_H_45_NO_7_S	+	Taurocholic acid	HMDB0000036	↓	Primary bile acid biosynthesis
32	10.97	398.3396	C_24_H_46_O_4_	+	Axillarenic acid	LMFA01050418	↓	Fatty acid metabolism
33	10.99	509.6566	C_25_H_52_NO_7_P	+	LysoPE (20:0)	HMDB0011511	↑	Glycerophospholipid metabolism
34	11.08	425.3505	C_25_H_47_NO_4_	+	Vaccenyl carnitine	HMDB0006351	↑	Fatty acid metabolism
35	11.69	523.3638	C_26_H_54_NO_7_P	+	PE (21:0/0:0)	LMGP02050026	↑	Glycerophospholipid metabolism
36	11.88	195.1721	C_9_H_9_NO_4_	+	3-hydroxyhippuric acid	HMDB0006116	↓	Glycerophospholipid metabolism
37	12.03	523.3638	C_26_H_54_NO_7_P	+	PC (18:0)	LMGP01050026	↓	Glycerophospholipid metabolism
38	12.03	273.1212	C_12_H_19_NO_6_	+	Glutaconylcarnitine	HMDB0013129	↓	Fat metabolism
39	12.78	120.1039	C_4_H_8_O_4_	+	4-deoxyerythronic acid	HMDB0000498	↑	Fatty acid metabolism

Note: ↑ or ↓ indicates that compared with the normal group, the metabolite content in the model group increases or decreases.

**Table 2 molecules-27-08860-t002:** Identification of the potential biomarkers in plasma of the MI rats based on ^1^H NMR.

No.	Potential Biomarkers	^1^H NMR
1	Leucine	δ 0.92 (d)
2	Valine	*δ* 1.04 (d), 3.61 (d)
3	Lactate	*δ* 1.35 (d)
4	Threonine	*δ* 1.36 (d), 3.58 (d)
5	Alanine	*δ* 1.48 (d)
6	Acetate	*δ* 1.94 (s)
7	Methionine	*δ* 2.16 (s)
8	Pyruvate	*δ* 2.34 (s)
9	Succinate	*δ* 2.40 (s)
10	3-Hydroxybutyrate	*δ* 1.20 (d)
11	Methylamine	*δ* 2.61 (s)
12	Dimethylamine	2.72 (s)
13	Trimethylamine	*δ* 2.96 (s)
14	Creatine	*δ* 3.04 (s)
15	Malonate	*δ* 3.12 (s)
16	*β*-glucose	*δ* 3.24 (d)
17	Glycine	*δ* 3.52 (s)

**Table 3 molecules-27-08860-t003:** Amelioration of different groups for the potential biomarkers related to MI.

No.	Identification	HMDB IDs	Normal vs. Model	Positive vs. Model	CNS vs. Model	NS vs. Model	SF vs. Model
1	8-Hydroxyguanosine	HMDB0002044	Δ *	Δ *	/	Δ *	/
2	Citrate	HMDB0000094	Δ **	Δ **	Δ **	/	Δ **
3	3-Oxocholic acid	HMDB0000502	Δ **	Δ **	Δ **	Δ **	Δ **
4	Ursocholic acid	HMDB0000917	Δ **	/	Δ **	Δ *	Δ *
5	Cholic acid	HMDB0000619	Δ **	Δ *	Δ *	Δ *	/
6	Deoxycholic acid	HMDB0000626	Δ **	Δ **	Δ *	/	Δ *
7	Phosphoglycolic acid	HMDB0000816	Δ *	/	/	/	Δ *
8	Succinate	HMDB0000254	Δ **	Δ **	Δ **	/	Δ **
9	alpha-Licanic acid	LMFA02000194	Δ *	Δ *	Δ *	Δ *	/
10	Malic acid	HMDB0000744	Δ **	/	Δ **	Δ **	/
11	Hydroxypropionylcarnitine	HMDB0013125	Δ **	/	Δ *	/	Δ *
12	3-Hydroxyundecanoic acid	HMDB0061654	Δ *	/	/	/	Δ *
13	Saccharopine	HMDB0000279	Δ *	/	/	/	/
14	PI (16:1/0:0)	LMGP06050009	Δ *	Δ *	/	/	/
15	Margaric acid	LMFA01010048	Δ **	Δ **	Δ **	Δ **	Δ **
16	8-Hydroxyguanine	HMDB0002032	Δ **	Δ **	Δ **	Δ **	/
17	LysoPC (20:5)	HMDB0010397	Δ *	/	/	/	/
18	Sphinganine	HMDB0000269	Δ **	Δ **	Δ **	Δ *	Δ *
19	Sphingosine 1-phosphate	HMDB0000277	Δ **	Δ **	Δ **	/	/
20	LysoPC (18:0)	HMDB0010384	Δ **	Δ **	Δ **	/	/
21	Sphingosine	HMDB0000252	Δ **	Δ **	Δ **	/	/
22	PE (19:0/0:0)	LMGP02050028	Δ **	/	Δ **	Δ **	Δ **
23	3-Hydroxynonanoyl carnitine	HMDB0061635	Δ *	/	/	/	/
24	Chenodeoxycholic acid	HMDB0000518	Δ **	Δ **	Δ **	/	/
25	PC (16:0/0:0)	LMGP01050018	Δ **	Δ **	Δ **	Δ **	Δ **
26	LysoPE (22:5/0:0)	HMDB0011524	Δ *	/	Δ *	/	/
27	PC (18:1/0:0)	LMGP01050029	Δ *	Δ *	/	/	/
28	LysoPC (18:1)	HMDB0002815	Δ *	Δ *	/	Δ *	/
29	Adrenoyl ethanolamide	HMDB0013626	Δ *	/	/	/	/
30	Uridine	HMDB0000296	Δ **	Δ **	Δ **	Δ **	Δ *
31	Taurocholic acid	HMDB0000036	Δ **	Δ *	Δ **	Δ *	Δ **
32	Axillarenic acid	LMFA01050418	Δ **	Δ **	Δ **	Δ **	/
33	LysoPE (20:0)	HMDB0011511	Δ *	Δ *	/	/	/
34	Vaccenyl carnitine	HMDB0006351	Δ **	/	Δ **	Δ **	/
35	PE (21:0/0:0)	LMGP02050026	Δ **	Δ **	Δ **	Δ **	Δ **
36	3-Hydroxyhippuric acid	HMDB0006116	Δ **	/	Δ *	/	/
37	PC (18:0)	LMGP01050026	Δ **	Δ **	Δ **	Δ **	Δ **
38	Glutaconylcarnitine	HMDB0013129	Δ *	Δ *	Δ *	/	Δ *
39	4-Deoxyerythronic acid	HMDB0000498	Δ **	/	Δ *	/	Δ *
40	*L*-Leucine	HMDB0000687	Δ **	Δ **	Δ **	Δ *	Δ **
41	*L*-Valine	HMDB0000883	Δ **	/	Δ **	/	/
42	Lactate	HMDB0000190	Δ **	Δ **	Δ **	Δ *	Δ *
43	Threonine	HMDB0000167	Δ **	Δ **	Δ **	Δ **	Δ *
44	Alanine	HMDB0000161	Δ *	/	/	/	/
45	Acetate	HMDB0000042	Δ *	/	/	/	/
46	Methionine	HMDB0000696	Δ *	/	/	/	/
47	Pyruvate	HMDB0000243	Δ **	Δ **	Δ **	/	Δ *
48	Succinate	HMDB0000254	Δ **	Δ **	Δ **	/	Δ *
49	3-Hydroxybutyrate	HMDB0000357	Δ *	/	/	/	/
50	Methylamine	HMDB0000164	Δ **	Δ **	Δ **	Δ **	Δ **
51	Dimethylamine	HMDB0000087	Δ **	Δ **	Δ **	Δ **	/
52	Trimethylamine	HMDB0000906	Δ **	Δ **	/	Δ **	/
53	Creatinem	HMDB0000064	Δ **	/	Δ **	/	/
54	Malonate	HMDB0000691	Δ **	Δ **	Δ **	Δ *	/
55	*β*-Glucose	HMDB0000122	Δ **	Δ **	Δ **	/	Δ *
56	Glycine	HMDB0000123	Δ **	Δ **	Δ **	/	/

Note: ∆ means AUC ≥ 0.8, / means AUC < 0.8; * *p* < 0.05 and ** *p* < 0.01.

## Data Availability

Not applicable.

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
