# Peer review of "The Synergistic Mechanism of Total Saponins and Flavonoids in Notoginseng–Safflower against Myocardial Infarction Using a Comprehensive Metabolomics Strategy"

_molecules, 2022, doi:10.3390/molecules27248860_

Round 1

Reviewer 1 Report

These authors investigated the therapeutic effect of Notoginseng-Safflower (NS-SF) on myocardial infarction rat and the synergistic action of NS and SF based on metabolomics techniques. They found that NS-SF could regulate the significantly regulate several metabolism pathways in MI rats and NS-SF showed a better pharmacological effects on NS or SF alone.

(1)         The pathogenesis of myocardial infarction needs to be described well. The authors found that NS-SF could regulate the disturbed energy metabolism, amino acid metabolism, glycerolipid metabolism, etc. What is the relationship of these metabolism pathways to myocardial infarction?

(2)         In the “materials and reagents”, please describe the preparation of NF and SF in detail because the preparation methods influence the constituents of NF and SF.

(3)       Even if no PK study was performed, the compounds/constituents that were most likely to be responsible for the pharmacological activities of NF and SF are suggested to be discussed, based on the current literature studies.

Author Response

(1) The pathogenesis of myocardial infarction needs to be described well. The authors found that NS-SF could regulate the disturbed energy metabolism, amino acid metabolism, glycerolipid metabolism, etc. What is the relationship of these metabolism pathways to myocardial infarction?

Reply 1:

Thank you for your comments. The relevant contents have been described in section of “3. Discussion”, for example, as follows:

Many studies have shown abnormal energy metabolism in MI disease states [16]. Pyruvate was not only the product of the glycolysis pathway but also the carboxylation of pyruvate, which was an essential physiological process to maintain normal heart function [17]. The tricarboxylic acid cycle played a central role in oxidative phosphorylation of the myocardium, so under normal conditions, the contents of tricarboxylic acid intermediates such as malic acid and succinic acid were strictly regulated by the body [18]. In this study, compared with sham rats, the plasma contents of pyruvate, acetate, glucose, succinate, citrate and malate, which are related to energy metabolism, were changed in MI rats, which suggested that MI mainly consumed glucose to obtain energy through glycolysis. The metabolites related to energy metabolism can be manipulated by NS, SF or NS-SF, and NS-SF could regulate succinate, citrate and glucose compared with NS and malic acid compared with SF.

(2) In the “materials and reagents”, please describe the preparation of NF and SF in detail because the preparation methods influence the constituents of NF and SF.

Reply 2:

Thank you for your comments. The contents have been added in section of “4.1 Materials and Reagents. The detailed contents are listed as follows:

NS, the total saponins of notoginseng, was purchased from Yunnan Plant Pharmaceutical Co., Ltd. (Kunming, China) and the quality standard of NS complied with Chinese Pharmacopoeia (2010 edition), with notoginsenoside R1, ginsenoside Rg1, ginsenoside Re, ginsenoside Rb1 and ginsenoside Rd contents of 6.2%, 26.6%, 4.1%, 32.5% and 6.6%, respectively. Safflower was collected from Xinjiang Uygur Autonomous Region (China) and authenticated by one of the authors, Prof. Pengfei Tu, as the flowers of Carthamus tinctorius L. The voucher specimen of safflower (No. 20110301) was deposited at the Modern Research Center for Traditional Chinese Medicine, Peking University (Beijing, China). The reflux procedures of the materials of safflower were 120 L reflux for 1 hour, 100 L reflux for 0.5 hour, and 100 L reflux for 0.5 hour at 80 ℃ using deionized water, and then the extract was filtered and concentrated in vacuo. The concentrated solution was subjected to a D101 macroporous resin column eluted with deionized water and 50% aqueous ethanol. SF was obtained by drying 50% aqueous ethanol eluate through spray drying. The levels of the effective components (total safflor yellow and total flavonoids) in SF were more than 40% using ultraviolet-visible spectrophotometry. Meanwhile, the content of hydroxysafflor yellow A and kaempferol-3-O-rutinoside in SF were more than 8.0% and 0.20% by high performance liquid chromatography, respectively [34]. The ratio of NS and SF was 6:5 in CNS.

(3) Even if no PK study was performed, the compounds/constituents that were most likely to be responsible for the pharmacological activities of NF and SF are suggested to be discussed, based on the current literature studies.

Reply 3:

Thank you for your comments. The contents have been added in section of “3. Discussion”. The detailed contents are listed as follows:

The saponins of notoginseng could significantly reduce the myocardial infarction area of LADCA model rats, improve the left ventricular diastolic and systolic functions, reduce the levels of creatine kinase and lactate dehydrogenase in serum, thus playing a role in myocardial protection [12]. Ginsenoside Rb1 and Rg1 were the main components of the saponins of notoginseng that showed as lowering blood pressure by regulating phosphatidylinositol-3 kinase, protein serine/threonine kinase, endothelial nitric oxide synthase signal transduction pathway and the transport of L-arginine in endothelial cells, and increasing endothelial cell dependent vasodilation in rats [13]. HSYA, the main component of safflower, had a myocardial protective effect by reducing the area of myocardial infarction, increasing the activity of SOD, reducing the content of MDA, inhibiting the activity of endothelial nitric oxide synthase, and reducing the levels of NO and creatine kinase isoenzyme [14-15]. The pharmacological results demonstrated that NS-SF had a therapeutic effect through ameliorating myocardial damage, apoptosis, easing oxidative stress and anti-inflammation in this research.

  1. Han, S.Y.; Li, H.X.; Ma, X.; Zhang, K.; Ma, Z.Z.; Jiang, Y.; Tu, P.F. Evaluation of the anti-myocardial ischemia effect of indi-vidual and combined extracts of Panax notoginseng and Carthamus tinctorius in rats. J Ethnopharmacol. 2013, 145(3), 722-727. doi: 10.1016/j.jep.2012.11.036.
  2. Pan, C.; Huo, Y.; An, X.; Singh, G.; Chen, M.; Yang, Z.; Pu, J.; Li, J. Panax notoginseng and its components decreased hyper-tension via stimulation of endothelial-dependent vessel dilatation. Vascul Pharmacol. 2012, 56(3-4), 150-158. doi: 10.1016/j.vph.2011.12.006.
  3. Wang, T.; Fu, F.H.; Han, B.; Li, G.S.; Zhang, L.M.; Liu, K. Hydroxysafflor yellow A reduces myocardial infarction size after coronary artery ligation in rats. Pharm Biol. 2009, 47(5), 458-462. doi: 10.1080/13880200902822612
  4. Wang, C.; Ma, H.; Zhang, S.; Wang, Y.; Liu, J.; Xiao, X. Safflor yellow B suppresses pheochromocytoma cell (PC12) injury induced by oxidative stress via antioxidant system and Bcl-2 /Bax pathway. Naunyn Schmiedebergs Arch Pharmacol. 2009, 380(2), 135-142. doi: 10.1007/s00210-009-0424-x.

Reviewer 2 Report

1.      The figures 1-3 and 5 were not consistent with the figure captions, and not accordance with the descriptions in the manuscript, please check the figures carefully.

2.      How to determine the oral dose of NS, SF, NS-SF?

3.     Part 2.5, “NS, SF and NS-SF could significantly regulate 23, 23 and 38 differential metabolites, respectively” was not consistent with the numbers in Table 3, please check it.

4.     Line 184, “Metabolites coregulated by NS, SF, and NS-SF were 3-oxocholic acid,…”, but “15 margaric acidd3” was also the coregulated metabolites in table 3. Additionally, please the names of No.10 and 15 in Table 3.

5.     Line 187, “Three metabolites (8-hydroxyguanosine, 3-hydroxyhippuric acid and trimethylamine) were uniquely regulated by NS.” should be revised as “Three metabolites (8-hydroxyguanosine, lysoPC(18:1) and trimethylamine) were uniquely regulated by NS.”

6.     In figure 6, succinate was regulated in NS-SF and SF groups, but it was only regulated in in NS-SF group in table 3, please check it.

7.      Please revise “1H NMR” as “1H NMR” in figure 4, figure 5 and line 416.

Author Response

  1. The figures 1-3 and 5 were not consistent with the figure captions, and not accordance with the descriptions in the manuscript, please check the figures carefully.

Reply 1:

Thank you for your comments. We really confused the positions of figure 2 and figure 3, and changed the positions of figure 2 and figure 3. In addition, the legend of figure 5 has been revised and supplemented as follow:

Figure 5. The OPLS-DA of the plasma samples from sham and model rats. (A) positive mode by UPLC-QTOF/MS; (B) a combination plot of S-plot and VIP values in positive mode; (C) negative mode by UPLC-QTOF/MS; (D) a combination plot of S-plot and VIP values in negative mode; (E) 1H NMR; (F) a combination plot of S-plot and VIP values in 1H NMR. Sham: sham-operated group; Model: model group.

  1. How to determine the oral dose of NS, SF, NS-SF?

Reply 2:

Thank you for your comments. According to the previous study of our group, the optimal ratio and dose of notoginseng-safflower was 6:5. The anti-myocardial ischemia effect of medium and high dose of notoginseng-safflower was explored, and the optimal dose of notoginseng-safflower was determined to be 55 mg/kg. The content has been described in “4.3 MI model and Drug Administration” as follow:

Based on previous research, CNS group, NS group, SF group, and positive group were treated with CNS (NS:SF = 6:5, 55.0 mg/kg/d), NS (30.0 mg/kg/d), SF (25.0 mg/kg/d) and diltiazem hydrochloride (30.0 mg/kg/d) once a day for 7 days after surgery, respectively [40].

  1. Part 2.5, “NS, SF and NS-SF could significantly regulate 23, 23 and 38 differential metabolites, respectively” was not consistent with the numbers in Table 3, please check it.

Reply 3:

Thank you for your comments. The contents have been changed as follows:

“NS, SF and NS-SF could significantly regulate 23, 23 and 38 differential metabolites, respectively” had changed to:

“NS, SF and NS-SF could significantly regulate 25, 25 and 40 differential metabolites, respectively”

  1. Line 184, “Metabolites coregulated by NS, SF, and NS-SF were 3-oxocholic acid,…”, but “15 margaric acidd3” was also the coregulated metabolites in table 3. Additionally, please the names of No.10 and 15 in Table 3.

Reply 4:

Thank you for your comments. The metabolite has been added in the manuscript and the names of No.10 and 15 have been changed in Table 3 as follows:

1) In the section of “2.5 NS, SF, and NS-SF showed different characteristics in improving the differential metabolites related to MI”: Metabolites coregulated by NS, SF, and NS-SF were 3-oxocholic acid, ursocholic acid, margaric acid, uridine, taurocholic acid, sphinganine, phosphatidylethanolamine (PE) (19:0/0:0), PE (21:0/0:0), phosphatidylcholine (PC) (16:0/0:0), PC (18:0), L-leucine, lactate, threonine and methylamine.

2) In Table 3: “Malic acida” and “Margaric acidd3” have been changed to “Malic acid” and “Margaric acid”.

  1. Line 187, “Three metabolites (8-hydroxyguanosine, 3-hydroxyhippuric acid and trimethylamine) were uniquely regulated by NS.” should be revised as “Three metabolites (8-hydroxyguanosine, lysoPC(18:1) and trimethylamine) were uniquely regulated by NS.”

Reply 5:

Thank you for your comments. The contents have been changed as follows:

“Three metabolites (8-hydroxyguanosine, 3-hydroxyhippuric acid and trimethylamine) were uniquely regulated by NS.” have been changed to:

“Three metabolites (8-hydroxyguanosine, lysoPC(18:1) and trimethylamine) were uniquely regulated by NS.”

  1. In figure 6, succinate was regulated in NS-SF and SF groups, but it was only regulated in in NS-SF group in table 3, please check it.

Reply 6:

Thank you for your comments. We have confirmed that succinate was regulated in NS-SF and SF groups, and the corresponding content has been revised in Table 3.

  1. Please revise “1H NMR” as “1H NMR” in figure 4, figure 5 and line 416.

Reply 7:

Thank you for your comments. “1H NMR” have been changed to “1H NMR” in figure 4, figure 5 and line 416.

Reviewer 3 Report

I have completed the review of the manuscript and my comments are highlighted below. The present study addresses the investigation of synergistic mechanism of total saponins and flavonoids from Notoginseng and Safflower against effects of isoproterenol induced myocardial infarction. The authors used a metabolomics approach to understand the protective mechanisms of NS-SF. Although the study is well outlined, the findings are not clearly highlighted. In my view the authors should describe the properties or characteristics that makes the combined therapeutic effect of NS-SF on MI advantageous over other treatments.

Abstract

The main metabolic changes induced by NS-SF treatment on MI should be highlighted. The way it was described shows general findings and not specific aspects of NS-SF treatment.

Introduction

Page 2, line 50-51. “It is necessary to explore the therapeutic effect of NS-SF on MI to provide new effective therapeutic drugs for the clinic” Reading the introduction it seems this sentence was included without a proper context. I think this justification should be improved.

Results

I think there was a mistake with the citation of the figures in the results, in the main text and in the figure legends, “e.g”:

Page 2, line 73-77. Are the authors referring to the result in figure 2?

Page 3, line 88. Figure 3 or figure 2?

Page 3, line 120. Figure 2?

The legends of figures 2 and 3 are changed.

The images of the immunohistochemistry results in figure 3 are too small to see the changes with a good definition. The quality should be improved.

All figures and their legends should be revised.

Page 4, line 117. Is the methydopamine quantified in the study? I think the authors were referring to malondialdehyde (MDA) a lipid peroxidation marker and one of the final products of polyunsaturated fatty acid peroxidation in the cells.

Page 5, line 134. Which metabolites are most abundant and differentially identified by MS from NS-SF treatments?

In PCA score plot analysis what are the major metabolites identified in the NS-SF group? These metabolites can be included in Figure 4.

The legends of the x and y axes in Figure 4A and 4B are not clear and should be improved. Standardize the colors of the groups in Fig. 4A, 4B and 4C.

Page 7. The legend in Figure 5 has no correspondence with Figures 5A-5F. What do the blue and green dots on the vocano plots mean?

Discussion

Page 8, line 211-212. "Studies had shown that NS-SF had a significantly stronger effect on MI than NS and SF" Based on what study did the authors make this statement? Please include the reference of this statement.

Overall the discussion is good. I have only one consideration: As reported by the authors, one of the effects of MI is alteration of the redox balance and increase of oxidative stresses. It was not discussed through which mechanisms NS-SF exerted its effects in order to induce the increase of SOD and GSH-Px activity and reduce the levels of MDA. The authors should discuss these findings.

Materials and Methods

Page 10, line 300. Why was the 6:5 molar ratio used? Did the authors test other molar ratios of NS and SF? If yes, please include a brief explanation in M&M section.

Author Response

Abstract

The main metabolic changes induced by NS-SF treatment on MI should be highlighted. The way it was described shows general findings and not specific aspects of NS-SF treatment.

Reply 1:

Thank you for your comments. As the Reviewer suggested, the contents have been changed in Abstract as follows:

Notoginseng and Safflower are commonly used traditional Chinese medicines for benefiting qi and activating blood circulation. A previous study by our group showed that the compatibility of the effective components of total saponins of notoginseng (NS) and total flavonoids of safflower (SF), named NS-SF, had a preventive effect on isoproterenol induced myocardial infarction (MI) in rats. However, the therapeutic effect on MI and the synergistic mechanism of NS-SF are still unclear. Therefore, integrated metabolomics, combined with immunohistochemistry and other pharmacological methods, was used to systematically research the therapeutic effect of NS-SF on MI rats and the synergistic mechanism of NS and SF. Compared to NS and SF, the results demonstrated that NS-SF exhibited a significantly better role in ameliorating myocardial damage, apoptosis, easing oxidative stress and anti-inflammation. NS-SF showed a more significant regulatory effect on metabolites involved in sphingolipid metabolism, glycine, serine and threonine metabolism, primary bile acid biosynthesis, aminoacyl-tRNA biosynthesis and tricarboxylic acid cycle, such as sphingosine, lysophosphatidylcholine (18:0), lysophosphatidylethanolamine (22:5/0:0), chenodeoxycholic acid, L-valine, glycine, and succinate than NS or SF alone, indicating that NS and SF produced a synergistic effect on the treatment of MI. This study will provide a theoretical basis for the clinical development of NS-SF.

Introduction

Page 2, line 50-51. “It is necessary to explore the therapeutic effect of NS-SF on MI to provide new effective therapeutic drugs for the clinic” Reading the introduction it seems this sentence was included without a proper context. I think this justification should be improved.

Reply 2:

Thank you for your comments. As the Reviewer suggested, the relational contents have been added as follows:

Traditional Chinese medicines (TCMs) have been used to prevent and treat cardiovascular diseases for a long time [5]. Panax notoginseng (Burk.) F. H. Chen (Notoginseng) and Carthamus tinctorius L. (Safflower) in the form of an herb pair are commonly used in cardiovascular diseases. Total saponins are the main bioactive components in notoginseng (NS), and total flavonoids are the major bioactive ingredients in safflower (SF), which display protective effects against MI injury [6]. Previous studies have shown that the combination of NS and SF, named NS-SF, has a preventive effect on isoproterenol (ISO)-induced MI, and NS-SF was significantly better than that of NS and SF single drugs, which had a significant synergistic effect [7]. Compared with preventive drugs, it is more urgent to develop therapeutic drugs for MI in clinical practice, but the therapeutic effect on MI and the synergistic mechanism of NS-SF are still unclear. Therefore, it is necessary to explore the therapeutic effect of NS-SF on MI to provide new effective therapeutic drugs for the clinic.

Results

I think there was a mistake with the citation of the figures in the results, in the main text and in the figure legends, “e.g”:

Page 2, line 73-77. Are the authors referring to the result in figure 2?

Page 3, line 88. Figure 3 or figure 2?

Page 3, line 120. Figure 2?

The legends of figures 2 and 3 are changed.

Reply 3:

Thank you for your comments. We really confused the positions of figure 2 and figure 3, and changed the positions of figure 2 and figure 3.

The images of the immunohistochemistry results in figure 3 are too small to see the changes with a good definition. The quality should be improved. All figures and their legends should be revised.

Reply 4:

Thank you for your comments. Figure 3 (modified to Figure 2) and its legend have been revised as follows:

Figure 2. Effects of NS, SF and NS-SF on myocardial fibrosis and apoptosis induced by MI in rat heart tissues. (A) H&E staining of left ventricular (LV) tissue showed pathological and morphological changes in different groups (magnification, ×400). The necrotic myocardium was labeled with black arrow, and inflammatory cells are labeled with a yellow arrow. (B) The results of TUNEL staining in different groups. Arrows indicate apoptotic cardiomyocyte nuclei. (C) Apoptosis rate of myocardial cells in each group (mean ± SD, n = 6). (D) The serum levels of MI-associated biochemical markers of rats in different groups (mean ± SD, n = 6). #p < 0.05, ##p < 0.01, when versus sham group; *p < 0.05, **p < 0.01, when versus model group. Normal: normal group; Sham: sham-operated group; Model: model group; Positive group: positive control group; NS: total saponins of notoginseng group; SF: total flavonoids of safflower group; NS-SF: the compatibility of NS and SF group.

Page 4, line 117. Is the methydopamine quantified in the study? I think the authors were referring to malondialdehyde (MDA) a lipid peroxidation marker and one of the final products of polyunsaturated fatty acid peroxidation in the cells.

Reply 5:

Thank you for your comments. The content has been changed as follows:

“The level of methydopamine (MDA), the end product of membrane lipid peroxidation in myocardial cells, was significantly increased, and the activities of superoxide dismutase (SOD) and glutathione peroxidase (GSH-Px), enzymes related to oxidative stress, were significantly decreased in MI rats” has been changed to:

“The level of malondialdehyde (MDA), the end product of membrane lipid peroxidation in myocardial cells, was significantly increased, and the activities of superoxide dismutase (SOD) and glutathione peroxidase (GSH-Px), enzymes related to oxidative stress, were significantly decreased in MI rats”.

Page 5, line 134. Which metabolites are most abundant and differentially identified by MS from NS-SF treatments?

Reply 6:

Thank you for your comments. The contents have been described in the section of “2.5 NS, SF, and NS-SF showed different characteristics in improving the differential metabolites related to MI” as follows:

Metabolites coregulated by NS, SF, and NS-SF were 3-oxocholic acid, ursocholic acid, uridine, taurocholic acid, sphinganine, phosphatidylethanolamine (PE) (19:0/0:0), PE (21:0/0:0), phosphatidylcholine (PC) (16:0/0:0), PC (18:0), L-leucine, lactate, threonine and methylamine. Three metabolites (8-hydroxyguanosine, 3-hydroxyhippuric acid and trimethylamine) were uniquely regulated by NS. Phosphoglycolic acid and 3-hydroxyundecanoic acid were individually reversed by SF. Compared with NS and SF, NS-SF showed a more significant regulatory effect on metabolites involved in sphingolipid metabolism, glycine, serine and threonine metabolism, primary bile acid biosynthesis, aminoacyl-tRNA biosynthesis and tricarboxylic acid cycle (TCA) cycle, such as sphingosine, sphingosine 1-phosphate, lysophosphatidylcholine (lysoPC) (18:0), lysophosphatidylethanolamine (lysoPE) (22:5/0:0), chenodeoxycholic acid, L-valine, glycine, creatine and succinate. The metabolomics results gathered by UPLC/MS and 1H NMR suggested that the effect of NS-SF on MI was significantly better than that of NS or SF.

In PCA score plot analysis what are the major metabolites identified in the NS-SF group? These metabolites can be included in Figure 4. The legends of the x and y axes in Figure 4A and 4B are not clear and should be improved. Standardize the colors of the groups in Fig. 4A, 4B and 4C.

Reply 7:

Thank you for your comments. Figure 4 has been revised as follows:

Page 7. The legend in Figure 5 has no correspondence with Figures 5A-5F. What do the blue and green dots on the vocano plots mean?

Reply 8:

Thank you for your comments. The legend of Figure 5 has been added and revised as follows:

Figure 5. The OPLS-DA of the plasma samples from sham and model rats. (A) positive mode by UPLC-QTOF/MS; (B) a combination plot of S-plot and VIP values in positive mode; (C) negative mode by UPLC-QTOF/MS; (D) a combination plot of S-plot and VIP values in negative mode; (E) 1H NMR; (F) a combination plot of S-plot and VIP values in 1H NMR. Sham: sham-operated group; Model: model group.

Discussion

Page 8, line 211-212. "Studies had shown that NS-SF had a significantly stronger effect on MI than NS and SF" Based on what study did the authors make this statement? Please include the reference of this statement.

Reply 9:

Thank you for your comments. The relational contents have been added as follows:

In terms of the pharmacodynamics results, the efficacy of NS-SF including ameliorating myocardial damage, apoptosis, easing oxidative stress and anti-inflammation was significantly stronger than that of NS or SF alone. Moreover, UPLC/MS and 1H NMR metabolomic methods were applied to explore the therapeutic mechanisms and synergistic mechanisms of NS-SF against MI in rats. Compared with NS and SF, NS-SF could significantly improve the abnormality of energy metabolism, amino acid metabolism, glycerophospholipid metabolism and bile acid metabolism caused by MI to better exert the therapeutic effect of NS-SF on MI rats, indicating that NS-SF had the characteristics of synergism.

Overall the discussion is good. I have only one consideration: As reported by the authors, one of the effects of MI is alteration of the redox balance and increase of oxidative stresses. It was not discussed through which mechanisms NS-SF exerted its effects in order to induce the increase of SOD and GSH-Px activity and reduce the levels of MDA. The authors should discuss these findings.

Reply 10:

Thank you for your comments. The relational contents have been added as follows:

Previous studies had shown that the glutamate content was positively correlated with the size of MI [19]. Valine had been shown to reverse physiological changes caused by hypoxia and to provide cardiac protection during acute ischemia and hypoxia. Alanine was a nitrogen transporter produced by skeletal muscle [20]. Pyruvate can be converted into alanine through transamination to meet the energy requirements of some cardiomyocytes and is also the main energy supply pathway of cardio-myocytes [21]. L-tryptophan was closely associated with immune system activation and inflammation [22-24]. In this study, plasma glutamate was significantly increased in model group, while the plasma glutamate, arginine, valine, alanine and glutamine were significantly decreased, indicating an abnormal amino acid metabolic pathway in model group. NS-SF group was better than NS group at regulating leucine and more at regulating valine and glycine. Compared with SF, NS-SF could better regulate valine and glycine. Methionine has been proved to improve cell oxidative balance and interact with various ROS, and the increase of methionine is related to the increase of SOD and GSH Px activity [25]. Thus, we speculated that NS-SF alleviated MI-induced oxidative stress through reducing the level of methionine. In addition, the increase of BCAAs (valine, leucine and isoleucine) could lead to the accumulation of superoxide thus promoting the progress of heart failure [26-27]. The results demonstrated that NS-SF could synergistically regulate the catabolism of impaired BCAAs in MI rats and reduce the level of MDA.

  1. Dos Santos, L.M.; da Silva, T.M.; Azambuja, J.H.; Ramos, P.T.; Oliveira, P.S.; da Silveira, E.F.; Pedra, N.S.; Galdino, K.; do Couto, C.A.; Soares, M.S.; Tavares, R.G.; Spanevello, R.M.; Stefanello, F.M.; Braganhol, E. Methionine and methionine sul-foxide treatment induces M1/classical macrophage polarization and modulates oxidative stress and purinergic signaling pa-rameters. Mol Cell Biochem. 2017, 424(1-2), 69-78. doi: 10.1007/s11010-016-2843-6.
  2. Sun, H.; Olson, K.C.; Gao, C.; Prosdocimo, D.A.; Zhou, M.; Wang, Z.; Jeyaraj, D.; Youn, J.Y.; Ren, S.; Liu, Y.; Rau, C.D.; Shah, S.; Ilkayeva, O.; Gui, W.J.; William, N.S.; Wynn, R.M.; Newgard, C.B.; Cai, H.; Xiao, X.; Chuang, D.T.; Schulze, P.C.; Lynch, C.; Jain, M.K.; Wang, Y. Catabolic Defect of Branched-Chain Amino Acids Promotes Heart Failure. Circulation. 2016, 133(21), 2038-2049. doi: 10.1161/CIRCULATIONAHA.115.020226.
  3. Lopaschuk, G.D. Metabolic Modulators in Heart Disease: Past, Present, and Future. Can J Cardiol. 2017, 33(7), 838-849. doi: 10.1016/j.cjca.2016.12.013.

Materials and Methods

Page 10, line 300. Why was the 6:5 molar ratio used? Did the authors test other molar ratios of NS and SF? If yes, please include a brief explanation in M&M section.

Reply 11:

Thank you for your comments. According to the previous study of our group, the optimal ratio and dose of notoginseng-safflower was 6:5. The anti-myocardial ischemia effect of medium and high dose of notoginseng-safflower was explored, and the optimal dose of notoginseng-safflower was determined to be 55 mg/kg. The content has been described in “4.3 MI model and Drug Administration” as follow:

Based on previous research, CNS group, NS group, SF group, and positive group were treated with CNS (NS:SF = 6:5, 55.0 mg/kg/d), NS (30.0 mg/kg/d), SF (25.0 mg/kg/d) and diltiazem hydrochloride (30.0 mg/kg/d) once a day for 7 days after surgery, respectively [12].

Round 2

Reviewer 1 Report

The manuscript has been revised as request.

Author Response

Thank you very much for your guidance and approval.

Reviewer 3 Report

In this revised version the authors improved the manuscript, enriched the discussion, and made improvements throughout the document. In this sense responded most part of my initial concerns. In this revised version my only recommendation is highlight the main metabolites identified in the NS-SF x groups in the figure 4 (PCA) and also in the figure 5 (OPLS-DA analysis).

Spelling out the abbreviations "HSYA", line 232, and "BCAA", line 266.

Author Response

  1. In this revised version the authors improved the manuscript, enriched the discussion, and made improvements throughout the document. In this sense responded most part of my initial concerns. In this revised version my only recommendation is highlight the main metabolites identified in the NS-SF x groups in the figure 4 (PCA) and also in the figure 5 (OPLS-DA analysis).

Reply 1:

Thank you for your comments. The points in PCA and OPLS-DA represent the samples, and the main metabolites can’t be marked. Therefore, we have added the S-plot diagrams with marked the main metabolites, and uploaded these figures as the file of “Supporting information”. As the Reviewer suggested, the figures have been added as follows: 

Figure S1 The representative base peak intensity chromatograms of the rat plasma samples in ESI positive and negative mode by UPLC-QTOF/MS. A, positive mode; B, negative mode.

Figure S2 The OPLS-DA analysis of the plasma samples from NS-SF and model rats by UPLC-QTOF/MS analysis. A, OPLS-DA score plot of plasma samples; B, S-plot from OPLS-DA of plasma samples.

Figure S3 S-plot of the plasma samples from sham and model rats by UPLC-QTOF/MS analysis.

  1. Spelling out the abbreviations "HSYA", line 232, and "BCAA", line 266.

Reply 2:

Thank you for your comments. As the Reviewer suggested, the abbreviations have been added as follows:

2-1: “Hydroxy safflower yellow A (HSYA), the main component of safflower, had a myocardial protective effect by reducing the area of myocardial infarction, increasing the activity of SOD, reducing the content of MDA, inhibiting the activity of endothelial nitric oxide synthase, and reducing the levels of NO and creatine kinase isoenzyme [14-15]”.

2-2: “Thus, we speculated that NS-SF alleviated MI-induced oxidative stress through reducing the level of methionine. In addition, the increase of branched chain amino acids (BCAAs), including valine, leucine and isoleucine, could lead to the accumulation of superoxide thus promoting the progress of heart failure [26-27]”.
